# Childhood cancer survival in the highly vulnerable population of South Texas: A cohort study

Shenghui Wu[1]*, Yanning Liu[2], Melanie Williams[3], Christine Aguilar[4,5], Amelie G. Ramirez[6,7], Ruben Mesa[8], Gail E. Tomlinson[4,5,8]

1 Department of Public Health & Exercise Science, Appalachian State University, Boone, NC, United States of America, 2 Department of Psychology, University of Texas at Austin, Austin, TX, United States of America, 3 Cancer Epidemiology and Surveillance Branch, Texas Cancer Registry, Austin, TX, United States of America, 4 Department of Pediatrics, University of Texas Health San Antonio, San Antonio, TX, United States of America, 5 Greehey Children's Cancer Research Institute, University of Texas Health San Antonio, San Antonio, TX, United States of America, 6 Department of Population Health Sciences, University of Texas Health San Antonio, San Antonio, TX, United States of America, 7 Institute for Health Promotion Research, University of Texas Health San Antonio, San Antonio, TX, United States of America, 8 University of Texas Health San Antonio Mays Cancer Center, San Antonio, TX, United States of America

* wus@appstate.edu

**Data Availability Statement:** Data cannot be shared publicly because of legal privacy protection of cancer registry data. Protecting patient confidentiality is paramount to the Texas Cancer

## Abstract

This study examines childhood cancer survival rates and prognostic factors related to survival in the majority Hispanic population of South Texas. The population-based cohort study used Texas Cancer Registry data (1995–2017) to examine survival and prognostic factors. Cox proportional hazard models and Kaplan-Meier survival curves were used for survival analyses. The 5-year relative survival rate for 7,999 South Texas cancer patients diagnosed at 0–19 years was 80.3% for all races/ethnicities. Hispanic patients had statistically significant lower 5-year relative survival rates than non-Hispanic White (NHW) patients for male and female together diagnosed at age≥5 years. When comparing survival among Hispanic and NHW patients for the most common cancer, acute lymphocytic leukemia (ALL), the difference was most significant in the 15–19 years age range, with 47.7% Hispanic patients surviving at 5 years compared to 78.4% of NHW counterparts. The multivariable-adjusted analysis showed that males had statistically significant 13% increased mortality risk than females [hazard ratio (HR): 1.13, 95% confidence interval (CI):1.01–1.26] for all cancer types. Comparing to patients diagnosed at ages 1–4 years, patients diagnosed at age < 1 year (HR: 1.69, 95% CI: 1.36–2.09), at 10–14 year (HR: 1.42, 95% CI: 1.20–1.68), or at 15–19 years (HR: 1.40, 95% CI: 1.20–1.64) had significant increased mortality risk. Comparing to NHW patients, Hispanic patients showed 38% significantly increased mortality risk for all cancer types, 66% for ALL, and 52% for brain cancer. South Texas Hispanic patients had lower 5-year relative survival than NHW patients especially for ALL. Male gender, diagnosis at age<1 year or 10–19 years were also associated with decreased childhood cancer survival. Despite advances in treatment, Hispanic patients lag significantly behind NHW patients. Further cohort studies in South Texas are warranted to identify additional factors affecting survival and to develop interventional strategies.

Registry (TCR) and is required by state law and rules (Health and Safety Code, Section 82.009; Texas Administrative Code, Title 25, Part 1, Chapter 91, Subchapter A). Data requests for personal identifiers (e.g., name, date of birth, address) or information on geographic areas below the county level (e.g., ZIP code, census tract) must be approved by the Department of State Health Services (DSHS) Institutional Review Board (IRB) and the Research Executive Steering Committee (RESC) before the TCR can release the data. When applying to receive the data from the Texas Department of State Health Services (DSHS) for research, researchers are required to sign off on the following item in form HRP-301: "All data that directly or indirectly identifies a person will not be shared with any individual outside the research team, or any other entity, agency, institution, or firm." Per the DSHS IRB's webpage, "All DSHS data, except for suppressed publicly available aggregate data, is considered identifiable. In certain presentations of data, the combination of values can pinpoint just one or a few cases (e.g. a rare cancer in a small town). Even if the personal identifiable information (PII) is removed from the data set, the data are potentially identifiable. Therefore, all DSHS data are considered identifiable and should be treated as such." Furthermore, item #3 of the TCR Data Security and Confidentiality Agreement states, "The data will not be made available to any other individual, agency, institution, or firm and controls will be maintained to prevent unauthorized access." Texas Cancer Registry (Mail Code 1928) Texas Department of State Health Services PO Box 149347 Austin, TX 78714-9347 Email: CancerData@dshs.texas.gov

**Funding:** This project was partially supported by the NCI 5 P30CA054174 (Mesa R, Ramirez A, and Tomlinson G). The funders had no role in study design, data collection and analysis, decision to publish, or preparation of the manuscript.

**Competing interests:** The authors have declared that no competing interests exist.

## Introduction

The current population of childhood cancer survivors in the United States is estimated to be over half a million [1, 2]. Texas (TX) is the second most populous state in the US [3]. South TX, the 38-county area encompassing a large portion of Texas-Mexico border counties, has a population of more than 4 million and includes 69% Hispanics, primarily of Mexican ancestry [4], and 25% non-Hispanic Whites (NHW) [5]. The counties along the Texas-Mexico border include more than 90% Hispanics [6]. The population of South TX is largely medically under-served and understudied, having a lower per capita personal income, higher unemployment and poverty rates, higher number of people with little to no formal schooling, higher percentage of uninsured residents, less access to health care services, and a higher prevalence of obesity (30% vs. 23%) compared to the nation as a whole [5, 7, 8]; these characteristics may all uniquely impact cancer patients' prognosis and survival and suggest significant but potentially modifiable disparities.

In TX, more than 1,800 children under age 20 are diagnosed with cancer and almost 200 children with cancer die annually [9]. However, because of major treatment advances in recent decades, 84% of children with cancer in the U.S. now survive 5 years or more [10].

Published studies of population-based childhood cancer survival are mainly focusing on the populations with a low proportion of Hispanics [11, 12]. The childhood cancer survival rate in South TX, a region marked by multiple health disparities, has not been previously studied in detail. This study examines survival rates and prognostic factors for childhood cancer survival in South TX based on data from the Texas Cancer Registry (TCR) with the intent to further define existing challenges and gaps in progress.

## Materials and methods

### Research design

This proposed study is a retrospective cohort study based on de-identified limited-use data from the TCR [6]. The study did not require informed consent and was exempted from review by the Appalachian State University Institutional Review Board.

### Childhood cancer survival data

Survival data was obtained from the TCR [6]. The TCR is an identically-organized, population-based registry of all 254 TX counties and follows all standards and coding criteria of the Surveillance, Epidemiology, and End Results (SEER) dataset, including possession of the North American Association of Central Cancer Registries (NAACCR) Gold Certification [6]. Survival in months, vital status (alive or dead), and cause of death were selected for male and female residents of the TCR for the 38 counties comprising South TX.

### Classification of malignancies

Patients are identified according to the International Classification of Childhood Cancer Recode Third Edition, World Health Organization (ICCC3)/WHO 2008 Definition, a recoded variable provided by SEER that is based on site/histology [13]. Broad types of cancer are identified according to the Site Recode International Classification of Diseases for Oncology (ICD-O-3)/WHO 2008 Definition [14]. The broad types of cancer grouping based on the Site Recode ICD-O-3 Definition as well as the specific cancer types based on the ICCC coding are shown in S4 Table.

## Identification of childhood cancer relative survival (RS)

Survival was defined as the time from initial diagnosis to the time of death, with censoring at date of last contact or December 31, 2018, whichever came first. The TCR [6] collects death certificate information on dates and underlying cause of death from the state Vital Statistics and the National Death Index to ensure complete and accurate death information, including deaths which occur out of state. Relative survival is a net survival measure representing cancer survival in the absence of other causes of death. Five-year RS of childhood cancer was calculated by dividing the overall five-year survival after childhood cancer diagnosis by the five-year survival as observed in a similar population not diagnosed with childhood cancer.

## Classification of ethnicity and urban/rural residence

For all groups compared, ethnicity was defined using the NAACCR Hispanic/Latino Identification Algorithm, version 2.2.1 [6] which is the best practice guideline that all registries follow [15] although the agreement between cancer registry data and self-report data for Hispanic ethnicity was moderate [16]. Urban/rural residence was identified using the US Department of Agriculture 2003 Urban/Rural Continuum criteria [17]. Rural-Urban Continuum Codes form a classification scheme that distinguishes metropolitan counties by the population size of their metro area, and nonmetropolitan counties by degree of urbanization and adjacency to a metro area or areas. Metropolitan counties with continuum codes 1, 2, and 3 are designated urban and non-metropolitan counties with codes 4–9 are typically considered to be rural.

## Potential risk/protective factor and survival data

Data on date of diagnosis, age, gender, race/ethnicity, stage at diagnosis, type of therapy, survival months, and cause-specific death at the individual level were obtained from the TCR [6].

## Statistical analysis

SEER*Stat software v 8.3.6 (SEER*Stat, NCI) generated five-year RS rates in the South TX datasets (n = 7,999) using survival sessions (detailed individual data unavailable). Case listing sessions were used to generate de-identified individual cancer records which were used to examine the prognostic factors of childhood cancer survival (n = 5,865) including age at diagnosis, gender, and race/ethnicity. The three most common cancer types (ALL, brain cancers, and bone cancers) were analyzed separately. Descriptive group characteristics were used to summarize the data. Chi-square tests for categorical variables and Student t-tests for continuous variables were conducted to assess differences between groups. Cox proportional hazard models were used to determine the association between potential factors and survival months for childhood cancer patients in South TX, controlling for covariates. The comparisons of stage and broad types of therapy across cancer types are imprecise as there are many types of cancer in children, not all types are similarly staged, and all are treated differently. Therefore, the information on stage and treatment was shown in **Table 2** but not included in multivariable-adjusted analyses for all cancer types, but for the three most cancer types separately. Covariates included gender, race/ethnicity, age at diagnosis, urban/rural residence, disease stage, surgery, chemotherapy, and radiotherapy. Adjusted estimates of hazard ratios for each factor was obtained. Kaplan-Meier survival curves were constructed to visualize survival probability over time by different subgroups (gender, race/ethnicity, and age at diagnosis) and the log-rank test or Cox regression model was used to compare the significance of the curves. Tests of statistical significance were based on two-sided probability, and $P < 0.05$ was

considered statistically significant. Statistical modeling was performed by using SAS 9.4 (SAS Institute, Cary, NC).

## Results and discussion

Table 1 shows South Texas childhood cancer 5-year RS by gender and race/ethnicity from 1995–2017 based on survival sessions (detailed individual data unavailable). South TX had 7,999 patients with childhood cancer diagnosed at ages 0–19 years including 4,314 males and 3,685 females. Of the 7,999 patients, 5,380 (73.5%) were Hispanic Whites, 1,786 (22.3%) were NHW, 229 (2.9%) were Blacks, and 48 (0.9%) were Asians. The 5-year RS for patients diagnosed at 0–19 years was 80.3% for all races/ethnicities during 1995–2017 (male: 78.8% and female: 82%). Hispanic patients had statistically significant lower survival rates than NHW for male and female together for ages greater than 5 years. Hispanic males had lower survival compared to NHW males diagnosed at 15–19 and overall 0–19 years. Similarly, Hispanic females had lower survival comparing to NHW females diagnosed at 10–14, 15–19 and also overall 0–19 years. South TX had slightly more male childhood cancer diagnoses (4,314 vs. 3,685) as well as cancer survivors (3,399 vs. 3,021) compared to females. However, male childhood cancer patients had significantly lower survival rates at diagnosis ages of 15–19 years and overall diagnosis ages 0–19 years compared with female cancer patients for all races/ethnicities (75.4% vs. 84.2% for 15–19 years; 78.8% vs. 82.0% for 0–19 years), as well as for Hispanic patients (73.7% vs. 82.2% for 15–19 years; 77.5% vs. 80.4% for 0–19 years) and NHW (81.4% vs. 91.2% for 15–19 years; 83.0% vs. 87.3% for 0–19 years) analyzed separately. S1–S3 Tables show 5-year RS by gender and race/ethnicity from 1995–2017 for the three most common cancer types clinically with sufficient case (death) numbers for analysis (ALL, brain cancers, and bone tumors). Of the 7,999 evaluated childhood cancers in South TX there were 1,409 patients with ALL (survival rate: 77.6%), 935 patients with brain cancer (survival rate: 68.7%), and 362 patients with bone cancers (survival rate: 69.1%). These major diagnosis groups were analyzed separately.

Patients with ALL diagnosed at older ages had worse survival for all races/ethnicities; this was especially notable for Hispanic patients ($Ps < 0.05$ for most diagnosis age groups for Hispanics vs. NHW). Those diagnosed at 15–19 years had lowest survival rates (47.7%) among different age groups and significantly lower compared to NHW diagnosed at 15–19 years (78.4%). In peak age of 1 to 4 years, female Hispanic patients also had significantly lower leukemia survival rates compared to female NHW (89.5% vs. 97.5%). Similar comparisons were not possible to be made for blacks due to the small number (S1 Table).

Survival rates for cancers of the brain were also significantly lower in Hispanic patients (65.7%, SE 1.9) than in NHW patients (74.6%, SE 2.9) diagnosed at overall ages 0–19 years (S2 Table). Most bone tumors generally develop in children older than 5 years and increase in number after age 10 years. Bone cancer survival rates in Hispanic patients were significantly lower than in NHW females diagnosed at 15–19 (60.2%, SE 9.9 vs 100%, SE 0) and also overall 0–19 years (65.7%, SE 4.7 vs 84.3%, SE 6.5). Similar comparisons were not possible to be made for black patients due to the small number (S2 and S3 Tables).

The 5-year relative cancer survival rates significantly increased with year of diagnosis (each 5-year period) for both male and female NHW ($Ps < 0.05$) but not at statistically significant rates for male ($P = 0.16$) and female Hispanic patients ($P = 0.09$) (S1 Fig). The most recent time interval compared to the earliest time interval, however, was significant among Hispanic patients. The time trend for specific cancer types was not meaningful due to the limited numbers in the individual time periods.

**Table 1. South Texas childhood cancer 5-year relative survival in different gender and races/ethnicities, 1995–2017[a].**

| Diagnosis age and race/ethnicity | Male and female | | Male | | Female | |
|---|---|---|---|---|---|---|
| | N (Percentage, %) | Relative survival (SE, %) | N (Percentage, %) | Relative survival (SE, %) | N (Percentage, %) | Relative survival (SE, %) |
| **0–<1 year** | | | | | | |
| All Races | 440 (100) | 75.0 (2.1) | 234 (100) | 75.1 (2.9) | 206 (100) | 74.9 (3.1) |
| NHW | 85 (19.32) | 81.5 (4.3) | 45 (19.23) | 84.7 (5.6) | 40 (19.42) | 77.9 (6.7) |
| Hispanics | 336 (76.36) | 73.0 (2.5) | 176 (75.21) | 72.3 (3.4) | 160 (77.67) | 73.8 (3.5) |
| Blacks | 16 (3.64) | 82.4 (9.9) | 12 (5.13) | 76.1 (12.7) | 4 (1.94) | 100.0 (0.0) |
| **1–4 years** | | | | | | |
| All Races | 1,392 (100) | 83.1 (1.0) | 782 (100) | 82.9 (1.4) | 610 (100) | 83.4 (1.6) |
| NHW | 279 (20.04) | 82.5 (2.3) | 159 (20.33) | 81.8 (3.1) | 120 (19.67) | 83.5 (3.5) |
| Hispanics | 1,059 (76.08) | 83.0 (1.2) | 594 (75.96) | 82.6 (1.6) | 465 (76.23) | 83.6 (1.8) |
| Blacks | 34 (2.44) | 85.0 (6.2) | 18 (2.3) | 94.5 (5.4) | 16 (2.62) | 74.1 (11.2) |
| **5–9 years** | | | | | | |
| All Races | 1,122 (100) | 81.6 (1.2) | 630 (100) | 81.6 (1.6) | 492 (100) | 81.5 (1.8) |
| NHW | 212 (18.89) | 86.3 (2.4) | 107 (16.98) | 86.1 (3.5) | 105 (21.34) | 86.5 (3.4) |
| Hispanics | 873 (77.81) | 80.1 (1.4) | 507 (80.48) | 80.3 (1.8) | 366 (74.39) | 79.8 (2.2) |
| Blacks | 23 (2.05) | 87.0 (7.0) | 11 (1.75) | 100.0 (0.0) | 12 (2.44) | 75.0 (12.5) |
| **10–14 years** | | | | | | |
| All Races | 1,130 (100) | 78.1 (1.3) | 588 (100) | 78.6 (1.8) | 542 (100) | 77.6 (1.9) |
| NHW | 234 (20.71) | 83.6 (2.5) | 123 (20.92) | 80.8 (3.6) | 111 (20.48) | 86.6 (3.4) |
| Hispanics | 847 (74.96) | 76.7 (1.5) | 438 (74.49) | 78.3 (2.1) | 409 (75.46) | 75.0 (2.2) |
| Blacks | 32 (2.83) | 74.6 (7.8) | 16 (2.72) | 75.1 (10.8) | 16 (2.95) | 74.1 (11.2) |
| **15–19 years** | | | | | | |
| All Races | 1,725 (100) | 79.4 (1.0) | 941 (100) | 75.4 (1.5) | 784 (100) | 84.2 (1.3) |
| NHW | 396 (22.96) | 86.0 (1.8) | 204 (21.68) | 81.4 (2.9) | 192 (24.49) | 90.9 (2.1) |
| Hispanics | 1,255 (72.75) | 77.5 (1.2) | 699 (74.28) | 73.7 (1.7) | 556 (70.92) | 82.2 (1.7) |
| Blacks | 52 (3.01) | 73.7 (6.4) | 26 (2.76) | 69.2 (9.2) | 26 (3.32) | 77.9 (8.9) |
| **0–19 years** | | | | | | |
| All Races | 7,999 (100) | 80.3 (0.5) | 4,314 (100) | 78.8 (0.6) | 3,685 (100) | 82.0 (0.7) |
| NHW | 1,787 (22.34) | 85.1 (0.9) | 931 (21.58) | 83.0 (1.3) | 856 (23.23) | 87.3 (1.2) |
| Hispanics | 5,880 (73.51) | 78.8 (0.6) | 3,214 (74.5) | 77.5 (0.8) | 2,666 (72.35) | 80.4 (0.8) |
| Blacks | 229 (2.86) | 77.8 (2.8) | 116 (2.69) | 77.6 (4.0) | 113 (3.07) | 77.7 (4.1) |

[a] *P* values < 0.05 for the below comparisons: NHW vs. Hispanics (male and female: 5–9 years, 10–14 years, 15–19 years, and 0–19 years; male: 15–19 years and 0–19 years; female: 10–14 years, 15–19 years, and 0–19 years); male vs. female (all races/ethnicities, Hispanics and NHW: 15–19 years and 0–19 years). *P* values > 0.05 for all other comparisons. Survival rates for groups with other races were not calculated due to the small event number.

Overall, 5,865 patients diagnosed with childhood cancer between 1995 and 2017 in South Texas had sufficiently detailed individual data generated by case listing sessions to be eligible for the analysis of prognostic factors (**Tables 2** and **S4**). The median survival time was 91 months (interquartile range: 34 to 171 months), 122 (64–193) months for living survivors, and 18 (8–36) months for deceased patients. The median age at latest analysis available was 18.3 years (interquartile range 11.6 to 24.3 years) with a range of 0 to 42.9 years, 20 (13–26) years for living survivors, and 13 (5.8–18) years for deceased patients. Male patients accounted for 55%. The percentage of overall Hispanic patients was 75%, and NHW was 21%. Most of these patients had urban residence (88%), although survival rates between rural and urban residence were nearly identical. For almost one-half of all live and dead patients, insurance status was

**Table 2. Characteristics by the vital status among South Texas childhood cancer patients.**

| Characteristics | Vital status (number, percentage) | | | P value |
|---|---|---|---|---|
| | Dead (n = 1,338) | Alive (n = 4,527) | Total | |
| Gender | | | | 0.04 |
| Female | 574 (42.9) | 2,086 (46.08) | 2,660 | |
| Male | 764 (57.1) | 2,441 (53.92) | 3,205 | |
| Race/ethnicity | | | | 0.0002 |
| Hispanic | 1,062 (79.37) | 3,354 (74.09) | 4,416 | |
| Non-Hispanic White | 225 (16.82) | 991 (21.89) | 1,216 | |
| Others | 51 (3.81) | 182 (4.02) | 233 | |
| Urban/rural residence | | | | |
| Urban | 1,178 (88.04) | 4,008 (88.54) | 5,186 | 0.62 |
| Rural | 160 (11.96) | 519 (11.46) | 679 | |
| Insurance | | | | <0.0001 |
| Yes | 481 (35.95) | 2,430 (53.68) | 2,911 | |
| No | 32 (2.39) | 118 (2.61) | 150 | |
| Unknown | 825 (61.66) | 1,979 (43.72) | 2,804 | |
| Diagnosis age (years)[a] | | | | <0.0001 |
| <1 | 128 (9.57) | 319 (7.05) | 447 | |
| 1–4 | 259 (19.36) | 1,144 (25.27) | 1,403 | |
| 5–9 | 241 (18.01) | 889 (19.64) | 1,130 | |
| 10–14 | 285 (21.30) | 857 (18.93) | 1,142 | |
| 15–19 | 425 (31.76) | 1,318 (29.11) | 1,743 | |
| Current age groups (years)[b] | | | | |
| <18 | 973 (73.94) | 1,858 (41.04) | 2,831 | <0.0001 |
| 18–<25 | 298 (22.64) | 1,358 (30.00) | 1,656 | |
| 25–<30 | 26 (1.98) | 630 (13.92) | 656 | |
| 30–<35 | 15 (1.14) | 411 (9.08) | 426 | |
| 35–<40 | 3 (0.23) | 231 (5.10) | 234 | |
| ≥40 | 1 (0.08) | 39 (0.86) | 40 | |
| Stage[c] | | | | <0.0001 |
| Localized | 307 (23.82) | 1,433 (32.34) | 1,740 | |
| Regional extension | 96 (7.45) | 475 (10.72) | 571 | |
| Distant | 704 (54.62) | 1,922 (43.38) | 2,726 | |
| Unknown | 182 (14.12) | 601 (13.56) | 783 | |
| Treatment | | | | 0.25 |
| Yes | 1,187 (90.4) | 4,050 (90.97) | 5235 | |
| No | 126 (9.6) | 382 (8.58) | 530 | |
| Surgery | | | | <0.0001 |
| Yes | 929 (69.43) | 3,448 (76.18) | 4,377 | |
| No | 409 (30.57) | 1,078 (23.82) | 1487 | |
| Chemotherapy | | | | <0.0001 |
| Yes | 945 (70.73) | 2,541 (56.28) | 3,486 | |
| No | 391 (29.27) | 1,974 (43.72) | 2,365 | |
| Radiation therapy | | | | 0.03 |
| Yes | 104 (7.80) | 277 (6.14) | 381 | |
| No | 1,229 (92.20) | 4,238 (93.86) | 5,467 | |

[a] $P>0.05$ for diagnosis age < 1 year vs. ages at 10–14 years, diagnosis age <1 year vs. ages at 15–19 years, diagnosis ages at 10–14 years vs. ages at 15–19 years, and diagnosis age at 1–4 years vs. ages at 5–9 years; $P<0.05$ for other two group comparisons.

[b] $P>0.05$ for current age 25–<30 years vs. 30–<35 years, current age 25–<30 years vs. 35–<40 years, current age 25–<30 years vs. ≥40 years, current age 35–<35 years vs. ≥40 years, current age 35–<40 years vs. ≥40 years; $P<0.05$ for any other two group comparisons.

[c] $P>0.05$ for stage as "localized" vs. "regional extension"; $P<0.05$ for stage as "distant" vs. "localized" and "distant" vs. "regional extension".

not available (48%), while 49.6% were confirmed to have insurance. At the end of follow-up, 4,527 (77.2%) patients were alive, and 1,338 (22.8%) were deceased. Cancer-specific death was 1,118 (84%). Deceased childhood cancer patients were more likely to be male, Hispanic, or diagnosed at age younger than 1 year or at 10–19 years, have tumor stage as "distant", i.e., metastatic (all $Ps < 0.05$).

The results of the univariate and multivariable-adjusted Cox proportional hazards model analyses were displayed in **Table 3**. Univariate analysis showed that gender, diagnosis age, ethnicity/race were statistically significant prognostic factors of survival. As almost one-half patients did not provide insurance status, the Cox proportional hazards model analyses did not include insurance status. The multivariable-adjusted analysis results showed that males had statistically significant 13% increased mortality risk compared to females (HR = 1.13). Comparing to patients diagnosed at ages 1–4 years, patients diagnosed at age < 1 years (HR = 1.69), 10–14 years (HR = 1.42), and 15–19 years (HR = 1.40) had statistically significant decreased survival rates, while those diagnosed at ages 5–9 years did not have statistically significant different survival rates. When compared to the NHW group, the Hispanic patients

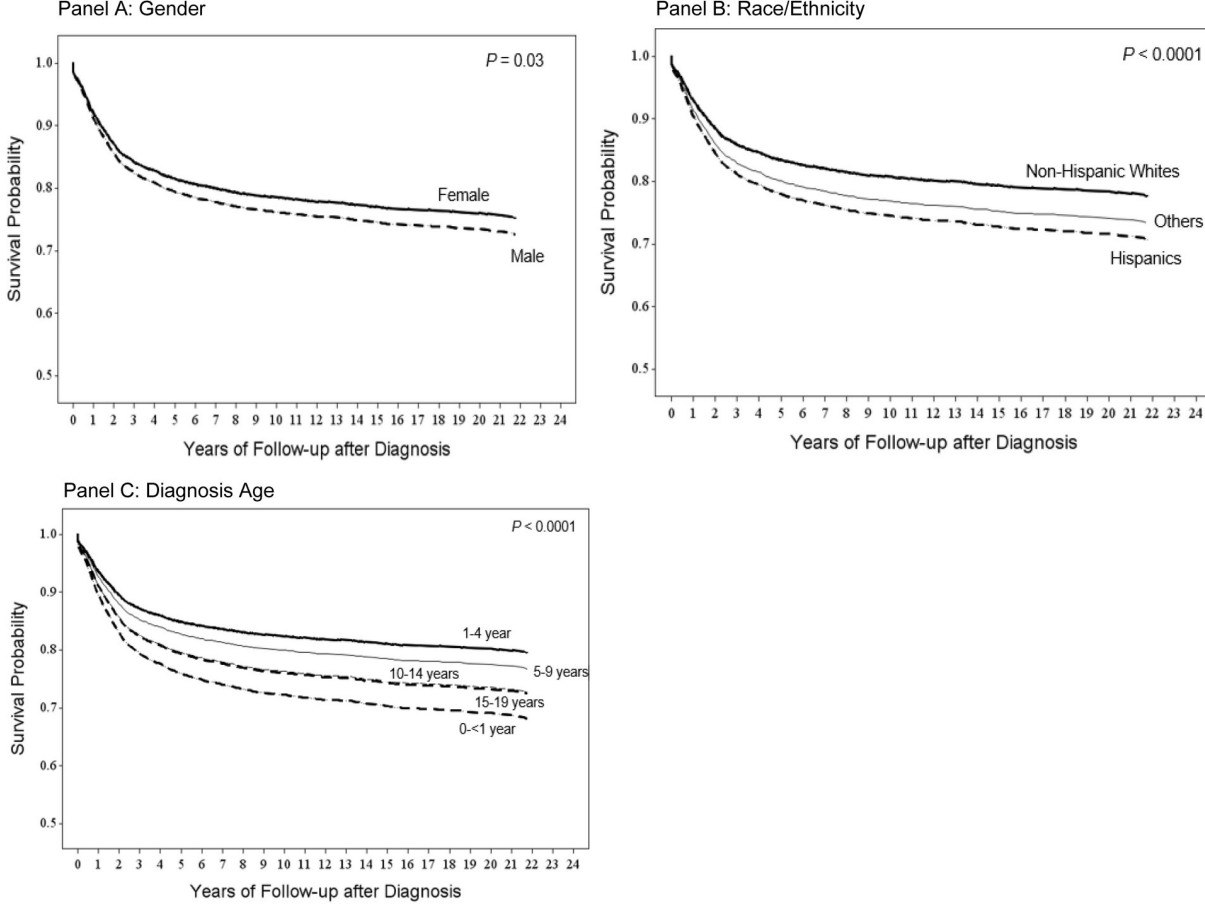

**Fig 1. South Texas childhood cancer multivariable-adjusted Kaplan-Meier survival curves by gender, race/ethnicity, and diagnosis age.** **(A)** Survival curves stratified by gender with the adjustment of race/ethnicity (2 groups: Hispanic and non-Hispanic whites) and diagnosis age (5 groups: <1 year, 1–4 years, 5–9 years, 10–14 years, and 15–19 years). **(B)** Survival curves in different races/ethnicities with the adjustment of gender (2 groups: male and female) and diagnosis age (5 groups: <1 year, 1–4 years, 5–9 years, 10–14 years, and 15–19 years). **(C)** Survival curves in different diagnosis age groups with the adjustment of gender (2 groups: Hispanic and non-Hispanic whites) and race/ethnicity (2 groups: Hispanic and non-Hispanic whites).

showed 38% increased mortality risk (HR = 1.38). **Fig 1A-1C** illustrate multivariable-adjusted Kapan-Meier survival curves for the different subgroups within overall patients. **Fig 1A** shows that male patients had statistically significant worse survival than female patients ($P$ = 0.03). **Fig 1B** displayed that Hispanic patients had significantly worse survival than patients from other race/ethnicity, and NHW had the best survival. **Fig 1C** showed that the survival probability was lower in the diagnosis ages < 1 year, 10–14, and 15–19 years, and highest in 1–4 and 5–9 years. It was not significantly different between patients diagnosed at 10–14 years and 15–19 years ($P$ = 0.91), between 1–4 years and 5–9 years ($P$ = 0.10), between 15–19 years and < 1 year ($P$ = 0.08), and between 10–14 years and < 1 year ($P$ = 0.11). The unadjusted Kapan-Meier survival curves were shown in the **S2A-S2C Fig**.

**Table 4** displayed multivariable-adjusted Cox proportional hazards model analyses for the most three cancer types. Comparing to patients diagnosed at ages 1–4 years, patients diagnosed at age < 1 years, 5–9 years, 10–14 years, and 15–19 years had statistically significant decreased ALL survival rates. Patients diagnosed at < 1 year had significant decreased brain cancer survival rates compared to ages 1–4 years. The survival probability was significantly lowest in ALL patients diagnosed at ages < 1 year and 15–19 years, followed by 10–14 years, then 5–9 years, and was highest in patients at ages 1–4 years. Comparing to the NHW, the Hispanic patients showed 66% increased ALL mortality risk, 52% increased brain cancer mortality risk, and 55% bone cancer mortality risk. Strikingly, comparing to the patients diagnosed at ages 1–4 years, those diagnosed at age < 1 year showed 608% increased ALL mortality risk and those diagnosed at ages 15–19 year showed 576% increased ALL mortality risk. The survival probability was significantly in brain cancer patients who received surgery or radiotherapy, and in bone cancer patients with tumor stage as "distant".

This study found that the 5-year RS for South TX cancer patients diagnosed at 0–19 years was 80.3% for all combined races/ethnicities during 1995–2017 (male: 78.8% and female: 82%). This was lower than national rates as the American Cancer Society reported that 84% of children with cancer survive 5 or more years [10]. Notable differences were observed for Hispanic and NHW patients. Hispanic patients had significant lower 5-year RS rates than NHW for male and female together diagnosed at 5 years of age and older. Male childhood cancer patients of all race and ethnicity groups had significantly lower survival rates at the combined diagnosis ages of 0–19 years and especially for 15–19 years, compared with females for all races/ethnicities together, as well as Hispanic and NHW patients analyzed separately. Survival trends over time were significantly increased for NHW but not for Hispanic patients, which lagged behind the increases seen in NHW patients. The multivariable-adjusted Cox proportional hazards model analysis showed that diagnoses age < 1 year or at 10–19 years, and Hispanic patients were associated with increased mortality risk/decreased childhood cancer survival rates compared to the corresponding counterparts. Compared to NHW, the Hispanic patients showed markedly increased mortality risk for the most three common cancers.

As described previously, the population of South TX is largely medically underserved from a socioeconomic perspective with high rates of poverty and lack of health insurance, low levels of education, and language limitation [5, 7, 8]. These factors may limit access to treatment and also to clinical trials and could also conceivably impact childhood cancer patients' prognosis and survival rates. Areas with the similar prevalence of above factors and other conditions as South Texas include US-Mexico border areas of California, Arizona, New Mexico South, and central and south areas of Florida [18]. Our findings may be generalizable to the above areas. Texas notably has higher rates of obesity compared with other areas of the country, with 30% of the population obese [5]. A recent study reported that pre-treatment obesity was associated with male and with Hispanic children with ALL [19]. These socioeconomic and behavioral factors might partly contribute to the differences in the national 5-year RS rates for the most

Table 3. Cox proportional survival analyses of South Texas childhood cancer patients.

| Covariates | Univariate Analysis | | Multivariable adjusted Analysis[a] | |
|---|---|---|---|---|
| | HR (95%CI) | P | HR (95%CI) | P |
| Gender (male vs. female) | 1.12 (1.01 to 1.25) | 0.04 | 1.13 (1.01 to 1.26) | 0.03 |
| Age at diagnosis (years) | | | | |
| < 1 vs. 1–4 | 1.68 (1.35 to 2.08) | <0.0001 | 1.69 (1.36 to 2.09) | < .0001 |
| 5–9 vs. 1–4 | 1.16 (0.97 to 1.39) | 0.10 | 1.16 (0.97 to 1.38) | 0.10 |
| 10–14 vs. 1–4 | 1.41 (1.19 to 1.67) | <0.0001 | 1.42 (1.20 to 1.68) | <0.0001 |
| 15–19 vs. 1–4 | 1.38 (1.18 to 1.61) | <0.0001 | 1.40 (1.20 to 1.64) | <0.0001 |
| Race/ethnicity | | | | |
| Hispanics vs. NHW | 1.37 (1.18 to 1.58) | <0.0001 | 1.38 (1.19 to 1.60) | 0.0002 |
| Others vs. NHW | 1.23 (0.91 to 1.67) | 0.18 | 1.23 (0.91 to 1.67) | 0.27 |
| Residence (rural vs. urban) | 1.05 (0.89 to 0.24) | 0.58 | 1.03 (0.87 to 1.21) | 0.77 |

[a] The results were generated with the adjustment of gender, race/ethnicity, urban/rural residence, and diagnosis age.

common children cancers as compared to our findings (90% [10] vs. 77.6% for ALL, and 74.7% [20] vs. 69% for brain cancer). The SEER data showed that the absolute inequality in 5-year cumulative incidence of ALL mortality in Hispanic patients changed from 10% (43% in Hispanic vs. 33% in NHW) in 1975–1983 to 7% (15% vs. 8%) in 2000–2010 [21], but Texas data were not included in the SEER program. A previous single institution study in South Texas reported lower survival outcomes of localized osteosarcoma in males compared to previously reported outcomes nationally [22]; the survival difference for bone tumors in national vs. South TX in our study was lower in South TX, albeit minimally (70% [23–25] vs. 69%) and the statistical significance is undetermined. However, here again, Hispanic patients with bone tumors had lower survival compared to NHW patients (66.5% in Hispanics and 77.4% in NHW).

As our study showed that the Hispanics' survival remains lower than that of NHW over the entire time course studied, it indicates that despite advances in treatment, there are still remaining disparities. The potential factors related to the disparities are unclear, however, importantly, we are clearly not "closing the gap" between Hispanic and NHW patients. Hispanic patients in South Texas are vulnerable to poverty-related health conditions and may lack health insurance or financial means to pay for quality health care and use fewer preventive care services than other ethnic groups [5, 18, 26], suggesting socioeconomic factor which could also contribute to worse survival rates in Hispanic comparing to NHW patients. Our study shows that variables including gender, diagnosis age, ethnicity, tumor stage, and treatment were associated with survival rates, however, many other data potentially influencing survival are not currently collected by the cancer registry. It has been shown that gender and lifestyle factors such as diet, physical activity, and age at diagnosis might affect childhood cancer survivors' health-related quality of life [27]. Adolescents with a history of cancer are at higher risk for developing smoking-related complications [28, 29], indicating an additional modifiable lifestyle factor possibly influencing survival. The incidence rates of ALL observed in South TX are higher than TX overall which is also higher than the U.S. overall [5]. Hispanic patients are known to have a significantly higher incidence of ALL and also worse survival than NHW and Asians [30]. The SEER program showed that Hispanic children and adolescents had somewhat poorer 5-year rates than NHW overall (74% vs. 81%; $P < 0.0001$) [31]. As described above, it was previously reported using SEER data that the absolute inequality in 5-year cumulative incidence of ALL mortality in Hispanic patients [21], but that study did not

**Table 4. Cox proportional survival multivariable-adjusted analyses[a] among the most three common South Texas childhood cancer diagnosis groups.**

| Covariates | Acute Lymphocytic Leukemia (n = 1,311) | | Brain Cancer (n = 819) | | Bone Cancer (n = 308) | |
|---|---|---|---|---|---|---|
| | HR (95%CI) | P | HR (95%CI) | P | HR (95%CI) | P |
| Gender (male vs. female) | 1.13 (0.80 to 1.44) | 0.33 | 1.03 (0.78 to 1.36) | 0.86 | 0.93 (0.59 to 1.47) | 0.77 |
| Age at diagnosis (years) | | | | | | |
| <1 vs. 1–4 | 7.08 (4.16 to 12.05) | < .0001 | 1.28 (0.70 to 2.32) | 0.43 | — | 0.99 |
| 5–9 vs. 1–4 | 1.51 (1.04 to 2.20) | 0.03 | 1.13 (0.77 to 1.65) | 0.55 | 3.80 (0.49 to 29.19) | 0.20 |
| 10–14 vs. 1–4 | 2.70 (1.86 to 3.90) | < .0001 | 1.13 (0.73 to 1.76) | 0.58 | 4.17 (0.56 to 31.24) | 0.16 |
| 15–19 vs. 1–4 | 6.76 (4.80 to 9.51) | < .0001 | 1.05 (0.67 to 1.64) | 0.84 | 4.88 (0.64 to 37.17) | 0.13 |
| Race/ethnicity | | | | | | |
| Hispanics vs. NHW | 1.66 (1.13 to 2.44) | 0.01 | 1.52 (1.05 to 2.20) | 0.03 | 1.55 (0.82 to 2.91) | 0.18 |
| Others vs. NHW | 1.36 (0.59 to 3.11) | 0.47 | 0.73 (0.26 to 2.07) | 0.55 | 1.71 (0.52 to 5.58) | 0.38 |
| Residence (rural vs. urban) | 0.88 (0.59 to 1.32) | 0.54 | 1.20 (0.79 to 1.82) | 0.38 | 1.30 (0.69 to 2.43) | 0.42 |
| Stage | | | | | | |
| Regional vs. localized | — | | 0.78 (0.48 to 1.28) | 0.33 | 1.76 (0.99 to 3.15) | 0.06 |
| Distant vs. localized | — | | 1.31 (0.82 to 2.08) | 0.26 | 2.83 (1.67 to 4.82) | 0.0001 |
| Surgery (yes vs. no) | — | | 0.47 (0.35 to 0.63) | <0.0001 | 0.76 (0.48 to 1.21) | 0.25 |
| Chemotherapy (yes vs. no) | 0.77 (0.51 to 1.18) | 0.24 | 2.61 (1.92 to 3.56) | <0.0001 | 1.29 (0.70 to 2.41) | 0.42 |
| Radiology (yes vs. no) | 1.42 (0.81 to 1.49) | 0.23 | 1.56 (1.10 to 2.21) | 0.01 | 0.82 (0.32 to 2.11) | 0.69 |

HR: hazard ratio, CI: confidence interval.

— No data was available due to very small number.

[a] The results were generated with the adjustment of gender, race/ethnicity, urban/rural residence, stage and diagnosis age.

[b] All patients with ALL had received surgery; the stage for almost all patients with ALL (99.9%) was regional, so the hazard ratio was not able to be estimated.

analyze the adolescent group (15–19 years) separately in which we see an even broader disparity persistent over time. The SEER program has not included Texas data with its high proportion of Hispanic patients, especially in South TX, where the population currently includes 69% Hispanic patients. In the analysis presented here, the disparity of survival of ALL is greatest overall with the difference most important in the 15–19 years old age range, an age range known to be associated with higher risk group subtypes [32]. The overall disparity in outcomes of adolescents with cancer was initially noted in 1996 [33] and has since been recognized as a need for a major increased effort in clinical trials [34].

The reasons for the extremely poor outcome in ALL in South Texas particularly for adolescents between 15 and 19 are not fully understood but undoubtedly involves the intersection of multiple factors. It has been reported that Hispanic patients from California with ALL, with similar ancestry, present with disease at older ages [35] which is a known risk factor overall. Genetic factors also play a role in cancer susceptibility and outcomes in Hispanic patients with variants in *ARID5B*, *IKZF3*, *CEBPE* and *CYP1A1* reported as contributory factors of risk in Hispanic patients [36–38]. *ARID5B* variants have also been linked to poor outcomes in Hispanic patients [39, 40].

This study is the first to examine childhood cancer survival and its prognostic factors in South TX that includes a majority proportion of Hispanic patients. This study has certain limitations. First, we were unable to examine other factors that could have affected the survival rate, such as socioeconomic conditions, insurance status, immigrant status, employment, education, access to care, social determinants of health, smoking, obesity, diet, exercise, posttreatment state-of-care, and pre-or post-diagnosis physical condition of the patient [12, 41–43]. Additional information on these potential modifiers and how they could interact with other prognostic factors for survival will require further cohort studies in South TX. Future research

will need to examine the intersections of contributions of molecular predisposition factors, response to treatment protocols especially in high-risk subgroups, lifestyle factors including those related to obesity, along with the impact of socioeconomic factors in underserved populations.

## Conclusions

Our study showed that Hispanic patients had statistically significant lower 5-year RS rates than NHW patients for both male and female in South TX, a Hispanic-majority region. Males had poorer survival compared to females for all races/ethnicities, as well as Hispanic and NHW patients analyzed separately. The disparities observed were largest for patients with ALL particularly for those diagnosed between 15 and 19. Those diagnosed at ages < 1 or at 10–19 years were significantly associated with decreased survival rates of childhood cancer compared to others. Hispanic patients showed an overall 38% increased mortality risk, and a 67% increased ALL mortality risk compared to NHW patients. Disparities persisted over the 22-year period studied, with Hispanic patients continuing to lag behind NHW in 5-year survival. To identify potential factors for intervention to improve survival, further cohort studies are warranted along with development of novel interventions.

## Supporting information

**S1 Fig. South Texas childhood cancer 5-year relative survival in different diagnosis years.** *P*-values for linear increasing trends of 5-year relative survival with year of diagnosis in different groups: Male NHW: *P* = 0.004; Female NHW: *P* = 0.004. Male Hispanics: *P* = 0.16; Female Hispanics: *P* = 0.09. The survival rates from female blacks showed an increasing trend from 1995 to 2017, however, the number for blacks is small and the trend was not statistically significant (*P* = 0.10).
(TIF)

**S2 Fig. South Texas childhood cancer unadjusted Kaplan-Meier survival curves by gender, race/ethnicity, and diagnosis age.**
(TIF)

**S1 Table. South Texas childhood acute lymphocytic leukemia 5-year relative survival in different gender and races/ethnicities, 1995–2017.**
(DOCX)

**S2 Table. South Texas childhood brain cancer 5-year relative survival in different gender and races/ethnicities, 1995–2017.**
(DOCX)

**S3 Table. South Texas childhood bone cancer 5-year relative survival in different gender and races/ethnicities, 1995–2017.**
(DOCX)

**S4 Table. Cancer types by the vital status among South Texas childhood cancer patients.**
(DOCX)

## Author Contributions

**Conceptualization:** Shenghui Wu.

**Data curation:** Shenghui Wu.

**Formal analysis:** Shenghui Wu.

**Funding acquisition:** Ruben Mesa.

**Investigation:** Shenghui Wu.

**Methodology:** Shenghui Wu, Yanning Liu.

**Project administration:** Shenghui Wu.

**Resources:** Shenghui Wu, Melanie Williams.

**Software:** Shenghui Wu.

**Supervision:** Shenghui Wu.

**Validation:** Shenghui Wu.

**Visualization:** Shenghui Wu.

**Writing – original draft:** Shenghui Wu.

**Writing – review & editing:** Shenghui Wu, Yanning Liu, Melanie Williams, Christine Aguilar, Amelie G. Ramirez, Ruben Mesa, Gail E. Tomlinson.

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
