## [Decision Letter · Decision Letter 0]

22 Dec 2022

PONE-D-22-31158Childhood Cancer Survival in the Highly Vulnerable Population of South Texas: A Cohort StudyPLOS ONE

Dear Dr. Wu,

Thank you for submitting your manuscript to PLOS ONE. After careful consideration, we feel that it has merit but does not fully meet PLOS ONE’s publication criteria as it currently stands. Therefore, we invite you to submit a revised version of the manuscript that addresses the points raised during the review process.

The manuscript needs some improvements and amendments to make any decision on the submission possible. Please follow the comments to refine the submission.

We look forward to receiving your revised manuscript.

Kind regards,

Sina Azadnajafabad, MD, MPH

Academic Editor

PLOS ONE

Journal Requirements:

Additional Editor Comments:

The authors may follow the PLOS One style guidelines to prepare the draft for revision.

The authors have provided too many tables and figures and they may consider moving some of them to the supplementary material accompanying the paper rather than including them in the main text. Also, Figure 2 contains a table, too.

Reviewers' comments:

Reviewer's Responses to Questions

**Comments to the Author**

1. Is the manuscript technically sound, and do the data support the conclusions?

Reviewer #1: Yes

Reviewer #2: Yes

2. Has the statistical analysis been performed appropriately and rigorously? 

Reviewer #1: Yes

Reviewer #2: Yes

3. Have the authors made all data underlying the findings in their manuscript fully available?

Reviewer #1: Yes

Reviewer #2: Yes

4. Is the manuscript presented in an intelligible fashion and written in standard English?

Reviewer #1: Yes

Reviewer #2: Yes

5. Review Comments to the Author

Reviewer #1: An interesting large-scale public health concern that has been addressed by authors in a scientific way. Though there are some analysis shortcomings, they are majorly from SEERs limitations which cannot be processed in this work. I have no comment or suggestion and would like to congratulate authors on their fascinating work.

Reviewer #2: - Can you please elaborate on the comment "uneducated inhabitants" to perhaps help quantify and clarify what that means for the reader

- Please clarify the rationale why insurance status could not be documented in 48% of subjects, this is central to understanding their access to care

- What percentage of these patients were undocumented immigrants without access to insurance?

- You analyze the survival based on different age groups, there is likely a component of survival distribution based on the types of cancers more commonly seen in these age groups, rather than the age itself being the factor of survival, but in fact its serving more of a surrogate for the types of cancers (some more or less aggressive) than others seen in the different patient age groups, can you please elaborate on this consideration.

- Besides the 'insurance status' which appears to be greatly lacking in 1/2 of the patients in this study, the authors do not provide data on 'access to care' or proximity to a medical center which could also serve as a surrogate to better understand the treatment options available to this vulnerable patient population.

- I appreciate the limitations stated by the authors: "...survival rate, such as

socioeconomic conditions, insurance status, employment, education, smoking, obesity, diet,

exercise, post-treatment state-of-care, and pre-or post-diagnosis physical condition of the patient" however, it seems that for a study such as this where the authors are examining the survival rates in a particularly medically underserved population, it seems that analyzing or obtaining these factors of 'social determinants of health' would be incredibly beneficial to better understand the dynamics truly affecting the survival disparities seen in this vulnerable patient population. This publication would be greatly enhanced were we able to obtain/analyze these important factors.

6. PLOS authors have the option to publish the peer review history of their article (what does this mean?). If published, this will include your full peer review and any attached files.

Reviewer #1: **Yes: **Esmaeil Mohammadi, MD MPH

Reviewer #2: No

---

## [Author Response · Author response to Decision Letter 0]

6 Feb 2023

Authors’ Responses to Reviewers’ Comments

RE: Childhood Cancer Survival in the Highly Vulnerable Population of South Texas: A Cohort Study 

Responses to Editor

1. Editor: Please ensure that your manuscript meets PLOS ONE's style requirements, including those for file naming. The PLOS ONE style templates can be found at 

Response: Thank you for your guidance. Please see the revised manuscript which meets PLOS ONE's style requirements.

2. Editor: We note that the grant information you provided in the ‘Funding Information’ and ‘Financial Disclosure’ sections do not match. 

Response: As suggested, the ‘Funding Information’ was checked and entered in the financial disclosure section of the submission system for the resubmission based on the submission guidelines. The funding ‘Financial Disclosure’ was included on the cover letter.

3. Editor: We note that you have indicated that data from this study are available upon request. PLOS only allows data to be available upon request if there are legal or ethical restrictions on sharing data publicly. For more information on unacceptable data access restrictions, please see http://journals.plos.org/plosone/s/data-availability#loc-unacceptable-data-access-restrictions. 

Response: As suggested, the data availability statement was updated and the data availability was addressed in the revised cover letter.

4. Editor: Your ethics statement should only appear in the Methods section of your manuscript. If your ethics statement is written in any section besides the Methods, please move it to the Methods section and delete it from any other section. Please ensure that your ethics statement is included in your manuscript, as the ethics statement entered into the online submission form will not be published alongside your manuscript. 

Response: As suggested, the ethics statement only appears in the Methods section of the revised manuscript. 

5. Editor: We note that Figure 1 in your submission contain [map/satellite] images which may be copyrighted. All PLOS content is published under the Creative Commons Attribution License (CC BY 4.0), which means that the manuscript, images, and Supporting Information files will be freely available online, and any third party is permitted to access, download, copy, distribute, and use these materials in any way, even commercially, with proper attribution. For these reasons, we cannot publish previously copyrighted maps or satellite images created using proprietary data, such as Google software (Google Maps, Street View, and Earth). For more information, see our copyright guidelines: http://journals.plos.org/plosone/s/licenses-and-copyright.

Response: Figure 1 was removed from the revised manuscript as suggested.

6. Editor: Please review your reference list to ensure that it is complete and correct. If you have cited papers that have been retracted, please include the rationale for doing so in the manuscript text, or remove these references and replace them with relevant current references. Any changes to the reference list should be mentioned in the rebuttal letter that accompanies your revised manuscript. If you need to cite a retracted article, indicate the article’s retracted status in the References list and also include a citation and full reference for the retraction notice.

Response: The reference list was reviewed as suggested, and it is complete and correct. There are no cited papers that have been retracted.

7. Editor: Additional Editor Comments:

The authors may follow the PLOS One style guidelines to prepare the draft for revision.

The authors have provided too many tables and figures and they may consider moving some of them to the supplementary material accompanying the paper rather than including them in the main text. Also, Figure 2 contains a table, too.

Response: Figure 1 was removed from the revised manuscript as suggested. Four tables and one figure were moved to the supplementary material. The table in Figure 2 (S1 Fig in the revised manuscript) was deleted. 

Responses to Reviewer 1

Reviewer: An interesting large-scale public health concern that has been addressed by authors in a scientific way. Though there are some analysis shortcomings, they are majorly from SEERs limitations which cannot be processed in this work. I have no comment or suggestion and would like to congratulate authors on their fascinating work.

Response: Thank you very much for your encouraging words. The limitations related to SEER database were addressed in the discussion section.

Responses to Reviewer 2

1. Reviewer: Can you please elaborate on the comment "uneducated inhabitants" to perhaps help quantify and clarify what that means for the reader

Response: As suggested, "uneducated inhabitants" was changed to “people with little to no formal schooling.”

2. Reviewer: Please clarify the rationale why insurance status could not be documented in 48% of subjects, this is central to understanding their access to care

Response: The SEER database could not provide the insurance status of these subjects. To make it clear, “insurance status could not be documented” was changed to “the insurance status was not available.”

3. Reviewer: What percentage of these patients were undocumented immigrants without access to insurance?

Response: Thank you for your suggestion. The immigrant status is not available from the SEER database, so the percentage of these patients who were undocumented immigrants without access to insurance was not available. This was described in the limitation section (paragraph 1 page 16).

3. Reviewer: You analyze the survival based on different age groups, there is likely a component of survival distribution based on the types of cancers more commonly seen in these age groups, rather than the age itself being the factor of survival, but in fact its serving more of a surrogate for the types of cancers (some more or less aggressive) than others seen in the different patient age groups, can you please elaborate on this consideration.

Response: Thank you for your suggestion. We analyzed cancer survival based on the age at diagnosis, not on the current age. Therefore, our results did not reflect the relationship between survival and current age. Patients diagnosed at age < 1 year and 10–19 years had a significant increase in mortality risk; in addition, we adjusted for age at diagnosis in other multivariable-adjusted analyses. 

4. Reviewer: Besides the 'insurance status' which appears to be greatly lacking in 1/2 of the patients in this study, the authors do not provide data on 'access to care' or proximity to a medical center which could also serve as a surrogate to better understand the treatment options available to this vulnerable patient population.

Response: Thank you for your suggestion. Data on 'access to care' is not available from the SEER database. It is one of the limitations and was described in the discussion section (paragraph 1 page 17).

5. Reviewer: I appreciate the limitations stated by the authors: "...survival rate, such as

socioeconomic conditions, insurance status, employment, education, smoking, obesity, diet, exercise, post-treatment state-of-care, and pre-or post-diagnosis physical condition of the patient" however, it seems that for a study such as this where the authors are examining the survival rates in a particularly medically underserved population, it seems that analyzing or obtaining these factors of 'social determinants of health' would be incredibly beneficial to better understand the dynamics truly affecting the survival disparities seen in this vulnerable patient population. This publication would be greatly enhanced were we able to obtain/analyze these important factors.

Response: Thank you for your suggestion. As suggested, 'social determinants of health' was added to this section (paragraph 1 page 17).

---

## [Decision Letter · Decision Letter 1]

15 Mar 2023

Childhood Cancer Survival in the Highly Vulnerable Population of South Texas: A Cohort Study

PONE-D-22-31158R1

Dear Dr. Wu,

We’re pleased to inform you that your manuscript has been judged scientifically suitable for publication and will be formally accepted for publication once it meets all outstanding technical requirements.

Kind regards,

Sina Azadnajafabad, MD, MPH

Academic Editor

PLOS ONE

Additional Editor Comments (optional):

Thanks for your efforts in revising the manuscript.

Reviewers' comments:

Reviewer's Responses to Questions

**Comments to the Author**

1. If the authors have adequately addressed your comments raised in a previous round of review and you feel that this manuscript is now acceptable for publication, you may indicate that here to bypass the “Comments to the Author” section, enter your conflict of interest statement in the “Confidential to Editor” section, and submit your "Accept" recommendation.

Reviewer #1: All comments have been addressed

Reviewer #2: All comments have been addressed

2. Is the manuscript technically sound, and do the data support the conclusions?

Reviewer #1: Yes

Reviewer #2: Yes

3. Has the statistical analysis been performed appropriately and rigorously? 

Reviewer #1: Yes

Reviewer #2: Yes

4. Have the authors made all data underlying the findings in their manuscript fully available?

Reviewer #1: Yes

Reviewer #2: Yes

5. Is the manuscript presented in an intelligible fashion and written in standard English?

Reviewer #1: Yes

Reviewer #2: Yes

6. Review Comments to the Author

Reviewer #1: (No Response)

Reviewer #2: Thank you for addressing all of the comments provided. The authors did a nice job of optimizing the manuscript.

7. PLOS authors have the option to publish the peer review history of their article (what does this mean?). If published, this will include your full peer review and any attached files.

Reviewer #1: **Yes: **Esmaeil Mohammadi, MD MPH

Reviewer #2: No

---

## [Editor Report · Acceptance letter]

27 Mar 2023

PONE-D-22-31158R1 

Childhood Cancer Survival in the Highly Vulnerable Population of South Texas: A Cohort Study 

Dear Dr. Wu:

I'm pleased to inform you that your manuscript has been deemed suitable for publication in PLOS ONE. Congratulations! Your manuscript is now with our production department. 

Kind regards, 

on behalf of

Dr. Sina Azadnajafabad 

Academic Editor

PLOS ONE